# Rat Model of Endogenous and Exogenous Hyperammonaemia Induced by Different Diets

**DOI:** 10.3390/ijms26051818

**Published:** 2025-02-20

**Authors:** Janine Donaldson, Tomasz Jacek, Piotr Wychowański, Kamil Zaworski, Dominika Szkopek, Jarosław Woliński, Danica Grujic, Stefan Pierzynowski, Kateryna Pierzynowska

**Affiliations:** 1School of Physiology, Faculty of Health Sciences, University of the Witwatersrand, Parktown, Johannesburg 2193, South Africa; 2Anara AB, 23132 Trelleborg, Sweden; dgrujic427@gmail.com (D.G.); stefan.pierzynowski@biol.lu.se (S.P.); 3National Research Institute of Animal Production, 32-083 Balice, Poland; tomasz.jacek@iz.edu.pl; 4Department of Head and Neck and Sensory Organs, Division of Oral Surgery and Implantology, Institute of Clinical Dentistry, Gemelli Foundation for the University Policlinic, Catholic University of the “Sacred Heart”, 00168 Rome, Italy; piotrwychowanski@wychowanski.pl; 5Department of Interventional Dentistry, Collegium Medicum, Nicolaus Copernicus University, 85-067 Bydgoszcz, Poland; 6Specialized Private Implantology Clinic Wychowański Stomatologia, 02-517 Warsaw, Poland; 7Department of Animal Physiology, The Kielanowski Institute of Animal Physiology and Nutrition, Polish Academy of Sciences, 05-110 Jabłonna, Poland; k.zaworski@ifzz.pl; 8Large Animal Models Laboratory, The Kielanowski Institute of Animal Physiology and Nutrition, Polish Academy of Sciences, 05-110 Jabłonna, Poland; d.szkopek@ifzz.pl (D.S.); j.wolinski@ifzz.pl (J.W.); 9Department of Biology, Lund University, 223 62 Lund, Sweden; 10Department of Medical Biology, Institute of Rural Health, 20-090 Lublin, Poland

**Keywords:** dietary hyperammonaemia, methionine–choline-deficient diet, ammonium acetate, rat model, hepatic encephalopathy

## Abstract

Two different diets able to induce dietary hyperammonaemia (a methionine–choline-deficient (MCD) diet and a methionine-deficient diet enriched with ammonium acetate (MAD + 20% ammonium acetate)) were tested in a rat model. The diets were shown to have different modes of action, inducing significant hyperammonaemia (HA) and growth retardation in the rats, with different metabolic consequences. The MCD diet resulted in the development of endogenous HA, with a decrease in bilirubin levels and an increase in hepatic fat content. In contrast, the MAD + 20% ammonium acetate diet increased circulating ALP and haptoglobin levels and decreased liver mass. The above results suggest that the MCD diet deteriorated the liver function of the rats, resulting in the development of endogenous HA, while the MAD diet caused moderate changes in liver metabolism, resulting in the development of exogenous HA. Interestingly, the commonly used oral treatments Lactulose and Rifaximin did not ameliorate hyperammonaemia during or after the treatment period. In conclusion, even though the diets used in the current study caused somewhat similar hyperammonaemia, they seemed to provoke different metabolic consequences. The latter can have an impact on the severity of the resulting hyperammonaemia and thus on the hyperammonaemia-induced encephalopathy, resulting in the development of distinguishing cognitive and metabolic (liver) effects compared to other forms of encephalopathy. We hypothesized that these rat models, with significantly increased serum ammonia levels, along with different liver injuries, could serve as a suitable double animal model for the testing of new, oral enzyme therapies for hepatic encephalopathy in future studies.

## 1. Introduction

Hyperammonaemia (HA) of any origin is toxic, especially for brain function since ammonium per se is the most effective poison for neurons. Hyperammonaemia related to a dietary-induced reduction in hepatic function, especially due to alcohol abuse and increased fat consumption, is most common. Hepatic encephalopathy (HE), resulting from hyperammonaemia, can range from mild cognitive and motor function disturbances to coma and even death in some instances [1]. Three types of HE have been described (Type A, B or C): Type A is related to acute liver failure and is characterised by the rapid development of hepatocyte necrosis and inflammation, with no pre-existing liver injury; Type B is related to portal–systemic shunting with no occurrence of liver disease/injury; and Type C is related to chronic liver disease (i.e., cirrhosis and portal–systemic shunting or portal hypertension) [2].

Animal models have provided us with substantial evidence and insight into the pathogenesis and pathophysiology of HA and HE and have been very useful for preclinical studies in the development of new therapeutic strategies/agents [2]. Various animal models have been used in HA studies including rats, mice, pigs, goats, rabbits, dogs and guinea pigs. Rats and mice are most used as HE models due to the low costs associated with the maintenance of the animals and the accessibility and availability of previous molecular, anatomical, behavioural, pathological and biochemical study results.

Elevation of serum ammonia has previously been induced with specific amino acid-deficient diets and supplementation with ammonia acetate [3,4]. Azorín et al. (1989) developed a simple rat model of hyperammonaemia more than 40 years ago [4]. Methionine–choline-deficient diets (MCD) have also previously been shown to result in hyperammonaemia in animal models of non-alcoholic fatty liver disease and non-alcoholic steatohepatitis [5].

The current study was designed primarily to highlight the metabolic specificity of endogenous HA related to HE, in comparison to dietary HA. Once hyperammonaemia was established in the rats, the efficacy of two short-term treatments (Lactulose and Rifaximin) to reduce serum ammonia was tested. The secondary aim of the study was to design and develop the most reliable and stable dietary hyperammonaemia rat model using specific diets for studying the neurodegenerative action of ammonium.

## 2. Results

### 2.1. Body Mass

There were no significant differences in body mass of the rats between groups at the beginning of the study (283 ± 6.81 g). Control rats grew during the weeks of the experimental period and were approximately 30% heavier than their initial body mass at the end of the five weeks (Figure 1). At the same time, the body mass of rats in both the MAD20 and MCD groups decreased during the first 8 weeks of the experimental period and then stabilised for the last 2 weeks, with their final body mass being approximately 25% lower than their initial body mass (Figure 1). The rats receiving the MAD20 diet exhibited a more significant drop in their body mass during the first 5 weeks of the experiment compared to the rats fed the MCD diet. There were no significant differences in body mass observed between rats in the MAD20 and MCD groups in the following weeks (week 6 to week 10) of the experimental period (*p* > 0.05).

Rats in both experimental groups showed a 50% reduction in their feed consumption compared to control rats kept on the standard diet.

### 2.2. Plasma Ammonia

The rats’ initial plasma ammonia levels were between 23 and 25 µg/dL and remained unchanged during the first five weeks of the experimental period (Table 1, Figure 2). A significant increase (10-fold) in plasma ammonia levels was observed in rats from both the MCD and MAD 20 groups during week 6 of the experimental period (Table 1, Figure 2). Plasma ammonia levels continued to increase significantly during the remaining four weeks of the experimental period and reached 800 µg/dL in the MAD20 rats and 895 µg/dL in the MCD rats at the end of the study, which was a 40-fold increase compared to initial levels (Table 1, Figure 2). The rise in blood ammonia levels was significantly higher in rats fed the MAD20 diet at all time points compared to rats fed the MCD diet. The plasma ammonia levels remained unchanged in the control rats during the four weeks of the experimental period.

The Lactulose/Rifaximin treatments did not have any significant effect on plasma ammonia levels (Table 1, Figure 2).

### 2.3. Plasma Alkaline Phosphatase (ALP) and Aspartate Aminotransferase (AST) Activity

Rats in the MAD20 group had significantly higher plasma ALP levels compared to those observed in rats in the MCD group (*p* = 0.0344, Figure 3) and rats in the control group (*p* = 0.0227, Figure 3). Plasma AST levels were similar in all groups (*p* > 0.05, Figure 4).

The ALP and AST levels were not affected by Lactulosum or Rifaximin treatment.

### 2.4. Plasma Bilirubin and Haptoglobin Levels

Plasma bilirubin levels were significantly lower in rats in the MCD group compared to those in the MAD20 group (*p* = 0.0042, Figure 5) and the control group (*p* = 0.0488, Figure 5). There were no significant differences in plasma bilirubin levels between control and MAD20 rats (*p* > 0.05, Figure 5). Rats in the MAD20 group had significantly higher plasma levels of haptoglobin compared to rats in the MCD group (*p* = 0.0195, Figure 6) and rats in the control group (*p* = 0.0240, Figure 6). Plasma haptoglobin levels were not different between the control and MCD rats (*p* > 0.05, Figure 6). The bilirubin and haptoglobin levels were not affected by Lactulosum or Rifaximin treatment.

### 2.5. Terminal Liver Mass and Liver Fat Content

Livers from rats in the MAD20 group were significantly lighter upon termination than those of rats from the MCD group (*p* = 0.0336, Figure 7). Terminal liver mass was not significantly different between rats in the control group and those in the MAD20 and MCD groups (*p* > 0.05, Figure 7). The fat content of livers from rats in the MCD group was significantly increased compared to that observed in both the control (*p* < 0.0001, Figure 8) and MAD20 rats (*p* < 0.0001, Figure 8). There was no significant difference in liver fat content between the control and MAD20 rats (*p* > 0.05). The liver mass and liver fat content were not affected by Lactulosum or Rifaximin treatment.

## 3. Discussion

The current study showed that both the MCD and MAD20 diets were successful in inducing chronic hyperammonaemia in the rats from week 6 until the end of the experimental period (week 10), as evidenced by the significantly increased plasma ammonia levels in the rats (a 40-fold increase compared to initial levels). The hepatic lipid content in the two chronic, dietary hyperammonaemia groups was significantly different, with exceptionally high liver lipid levels observed in the MCD rats, with endogenous HA. No changes in plasma ALP, AST or haptoglobin were observed in rats in the MCD group, with these parameters remaining at similar levels to those observed in the control rats. Plasma bilirubin levels in the MCD rats were significantly lower than those of rats in both the control and MAD20 groups. Rats in the MAD20 group had significantly increased plasma ALP and haptoglobin levels compared to the other two groups. Plasma AST and bilirubin levels, as well as liver lipid content, remained unchanged in the MAD20 group, with similar levels to those in the control group. The hyperammonaemia treatments assessed (Lactulosum and Rifaximin) had no significant effects on any of the investigated parameters.

Alanine aminotransferase (ALT) and aspartate aminotransferase (AST) are liver enzymes which catalyse the transfer of amino acids in the process of gluconeogenesis and amino acid metabolism [6,7]. Serum/plasma levels of ALT and AST are often used as markers of liver health [6]. Elevated levels of ALT and AST can be indicative of liver damage or inflammation and are often observed in patients with non-alcoholic fatty liver disease (NAFLD) [7]. Alkaline phosphatase (ALP) is a hydrolase enzyme which transports metabolites across cell membranes and is found in a variety of tissues including the bones, bile ducts, intestines and liver [7,8]. Circulating ALP levels are affected by changes in bone metabolism, biliary dysfunction and liver inflammation, with elevated levels associated with bone disorders and bile duct or liver problems [7,8]. Considering the unchanged plasma ALP and AST levels in the MCD rats, we can assume that the MCD diet did not result in any significant liver damage, despite the substantial increase in plasma ammonia levels and accumulation of lipids within the livers of these rats. Rats in the MAD20 group displayed increased plasma ALP levels together with unchanged plasma AST levels and no hepatic lipid accumulation. This may indicate that the raised plasma ALP levels observed in these rats might be from an extra-hepatic source. Harikrishnan et al. (2008) observed a significant increase in circulating ammonia, AST, ALT and ALP levels in male albino Wistar rats IP injected with ammonia chloride (100 mg/kg body mass), three times a week for eight consecutive weeks [9]. Gokcimen et al. (2007) observed no significant changes in serum AST and ALT levels in female Wistar rats IP injected with ammonium acetate (2.5 mmol/kg/day) for 28 days, even though significantly increased serum ammonia levels and some structural changes in the livers of the rats were observed [10]. The authors speculated that the minimal hepatic structural changes were accompanied by an increase in antioxidant defence systems; thus, no changes in the biochemical parameters were observed [10], which could also be the case with the MCD rats in the present study. Methionine and choline-deficient diets are commonly used in NAFLD animal studies. Methionine is an essential amino acid that cannot be synthesized de novo and is necessary for the synthesis of, among other things, phosphatidylcholine. Choline is an important component of cell membranes and is also required for phosphatidylcholine synthesis. Thus, a deficiency of both methionine and choline results in reduced synthesis of phosphatidylcholine, which in turn, adversely affects the assembly and secretion of very low-density lipoproteins (VLDLs) [5,11,12]. Without VLDLs, triglycerides cannot be effectively cleared from the hepatocytes and lipids accumulate within the liver, causing hepatic steatosis or NAFLD [13,14].

Previous studies have shown that even the early stage of NAFLD development (prior to the development of fibrosis and cirrhosis) is associated with reduced capacity of the liver for urea synthesis, with decreased expression and function of the enzymes involved in the urea cycle, including carbamoyl phosphate synthetase I (CPS1) and ornithine transcarbamylase (OTC) [15,16,17]. Since the urea cycle is responsible for the conversion of free ammonia (which is highly toxic) to non-toxic urea, which is then eventually excreted in the urine, any reduction in the liver’s capacity for urea synthesis results in ammonia accumulation and hyperammonaemia develops [15,16,17]. Recently, Mercado-Gomez et al. (2024) observed increased expression of hepatic glutaminase (GLS), which was associated with hepatic ammonia accumulation, in a choline-deficient, diet-induced animal model of metabolic-associated steatotic liver disease (MASLD), prior to any changes in urea cycle enzymes [18]. The ‘early’ increased expression of GLS was accompanied by the reduced expression of the glutaminase isoform, GLS 2. The ammonia synthesized by GLS is widely distributed throughout the liver parenchyma and thus could have a more significant damaging effect with regards to the hepatic accumulation of ammonia, compared to that which is synthesized by GLS2, which is confined to the periportal region of the liver and serves as an amplification system to increase ureagenesis and clear excess ammonia [18]. The authors speculated that the initial diet-induced increase in GLS expression, which was attributed to hepatic TLR-4 signaling, may result in changes in the methylation of the promoters of the urea cycle enzymes, as well as those of GLS and GLS2, downregulating and decreasing their expression, causing a further increase in hepatic ammonia and ultimately leading to the systemic hyperammonaemia associated with steatohepatitis [18].

The steatosis observed in the livers of the MCD rats, as evidenced by the significantly increased liver fat content (mg of fat per g of liver tissue), was not accompanied by an increase in terminal liver mass. It was, however, associated with a significantly reduced plasma bilirubin concentration. Bilirubin, the yellow bile pigment, is a breakdown product of haem proteins. Low levels of circulating bilirubin (hypobilirubinaemia) have been associated with the development of various cancers, cardiovascular disease, stroke, and colitis, as well as metabolic and cognitive disorders [19]. Previous studies have also shown that circulating bilirubin levels are inversely associated with NAFLD. Kwak et al. (2012) observed significantly lower levels of bilirubin in patients with ultrasonography-diagnosed NAFLD compared to those without NAFLD in a cross-sectional study of 17,348 participants [20]. Tian et al. (2016) found that conjugated bilirubin levels were significantly correlated with reduced risk of developing NAFLD, and the association was independent of classical risk factors such as coronary artery disease, diabetes, liver enzyme levels, and features of metabolic syndrome, as well as other standard metabolic risk factors [21]. More recent studies, however, have observed no independent relationship between serum bilirubin levels and NAFLD [22,23]. Thus, future studies should further explore the causal association between bilirubin and NAFLD, providing more insight into the reduced bilirubin levels and significant liver fat accumulation observed in the MCD rats of the current study. The normal plasma bilirubin and AST levels and the absence of hepatic steatosis in the MAD20 rats were also associated with significantly increased plasma haptoglobin levels. Haptoglobin is an acute phase protein synthesized in the liver. The primary function of haptoglobin is to bind haemoglobin released from erythrocytes during haemolysis, to prevent iron deficiency as well as the toxic effects of free haemoglobin [24]. In addition to its antioxidant function, haptoglobin also possesses anti-inflammatory properties and its synthesis is upregulated by pro-inflammatory cytokines including IL-1, IL-6 and TNF-α [25,26]. Thus, it seems that haptoglobin is upregulated in conditions of stress and inflammation [27], which may be why the plasma haptoglobin levels were significantly increased, along with the plasma ammonia levels, in the MAD20 rats in the current study.

Once the chronic hyperammonaemia was successfully induced with the high ammonia diets (MCD and MAD20) in the rats in the current study, two different treatments for hyperammonaemia, Lactulosum and Rifaximin, were tested. The main goal of such treatments is to reduce ammonia production, while at the same time maximizing the body’s capacity to remove ammonia from the bloodstream [28]. Strangely, neither of the two treatments administered to the rats were successful in reducing plasma ammonia levels or improving any of the hyperammonaemia-related changes observed in any of the other parameters assessed. In contrast, Jia and Zhang (2005) observed a significant reduction in ammonia levels, serum endotoxin levels, liver injury and the incidence of minimal hepatic encephalopathy (MHE) in male Sprague Dawley rats that received 8 mL/kg body mass of Lactulose, once a day for 8 days, following the induction of MHE by IP injection of thioactamide (200 mg/kg) [29]. Similarly, both Odena et al. (2012) and Tamaoki et al. (2016) observed significantly reduced blood ammonia levels in their Sprague Dawley rat models of hepatic encephalopathy (HE) that were treated with rifaximin (50 mg/kg daily by gavage for 14 days or 0.3, 3 and 30 mg/kg daily for 3 days) [30,31]. These differences in results obtained could be due to the different species of rats used, as well as the differences in the dosage and duration of treatments.

In summary, both diets (MCD and MAD20) induced massive hyperammonaemia and growth retardation in the rats, while seemingly having different modes of action and different metabolic consequences. The methionine–choline-deficient (MCD) diet resulted in the development of endogenous HA, with a decrease in bilirubin levels and an increase in hepatic fat content. In contrast, the methionine-deficient diet enriched with ammonium acetate (MAD + 20% ammonium acetate) increased circulating ALP and haptoglobin levels and decreased liver mass. The above results suggest that the MCD diet deteriorated the liver function of the rats, resulting in the development of endogenous HA, while the MAD20 diet caused moderate changes in liver metabolism, resulting in the development of exogenous HA. Interestingly, commonly used oral treatments, Lactulose and Rifaximin, did not ameliorate the hyperammonaemia or its metabolic features. The results of the present study should be interpreted in the context of the study limitations, including the lack of any molecular-based investigations (i.e., expression of GLS and urea cycle enzymes) concerning the mechanisms involved in the induction of the two seemingly different modes of action and the metabolic consequences of the MCD and MAD20 diets. Future studies should address these aspects, as well as include liver histological investigations and behavioural and cognitive analyses, which would provide more insight into the neurodegeneration induced by the diets.

## 4. Materials and Methods

### 4.1. Animals and Diets

Twenty-four, male Wistar rats, 8 weeks of age, weighing 283 ± 6.81 g (at the start of the study) were used in the current study and were generated at the Mossakowski Institute of Experimental Medicine, Polish Academy of Sciences, Warsaw, Poland. Procedures involving the care and the use of animals in this study were reviewed and approved by the II Local Ethical Commission in Warsaw (WAW2/120/2021) and were conducted in accordance with the principles outlined in the current Guide to the Care and Use of Experimental Animals.

The rats were maintained on a 12 h day–night cycle, with lights on from 6:00 am to 6:00 pm (06:00–18:00 h), at room temperature and a humidity of 30%. The rats were group-housed before the study commenced and then randomly allocated to cages (2–3 rats per cage) with beta chips as bedding. The rats were weighed and their body weight recorded prior to the randomization, at the end of the acclimatization period and then again at the end of each week during the experimental period. The rats had free access to feed and water for the duration of the study, during both the acclimatization and experimental periods. Feed and water intake was also recorded weekly during the experimental period. The chronic hyperammonaemia model was induced using two different custom-made, experimental diets/feed formulations (SSNIF, Soest, Nordrhein-Westfalen, Germany) [EF TD.90262 MCD w/o Met & Choline AA diet, #E15653-94 and EF TD.90262 mod. 20% Ammonia Acetate diet, #S5061-E010] and Control standard rat chow. The ingredients and proximate composition of the experimental diets, as well as the control diet/normal rat chow, are shown in Table 2. Cage-side clinical signs (ill health, behavioural changes, etc.) were also recorded once daily during all phases of the study.

### 4.2. Experimental Design

The rats underwent an adaptation period of 7 days, during which they were acclimated to the facility and the group housing, and they were fed a regular chow rat diet/control diet (Table 2). On the last day of the adaptation period, the rats’ body weight was measured and blood was collected for baseline serum ammonia measurements. The rats were then divided (based on body weight) into three cohorts: two special diet groups (MCD, n = 8 and MAD20, n = 8), and one control group, which was fed a regular chow rat diet, n = 4. Rats in the MCD group were fed a diet without methionine and choline and rats in the MAD20 group were fed a modified MCD diet supplemented with 20% ammonia acetate, for a period of 10 weeks (chronic hyperammonaemia models). Body weight and feed and water intake were measured weekly, and blood was collected once a week to check the plasma levels of ammonia.

### 4.3. Hyperammonaemia Treatments

Four weeks after the start of the chronic hyperammonaemia development on the customized diets/feed formulations, two treatments to reduce serum ammonia were tested. Four rats from each of the dietary groups (MAD20 and MCD groups) received Lactulose (Lactulosum, 10 g/kg bwt/day), while the remaining four rats from each group received Rifaximin (Xifaxan, 50 mg/kg bwt/day), administered twice daily via oral gavage for six days. After that, the treatment with either Lactulosum or Rifaximin continued for another five days for two rats from each experimental group (see Figure 9 detailing experimental design). All rats that received the Lactulosum and Rifaximin treatments continued for the duration of the experiment on their respective diets.

### 4.4. Blood Sampling and Analyses

Blood samples for ammonium estimation were collected into lithium heparin tubes, from the submandibular vein of all rats (control, MCD and MAD20 groups), under light isoflurane anaesthesia, at the end of the adaptation period and once a week during the experimental period. Upon termination, blood was withdrawn from the heart for plasma ammonia, alkaline phosphatase (ALP), aspartate aminotransferase (AST), bilirubin and haptoglobin measurements. The blood samples were stored at −20 °C until they were analysed. The blood samples were then centrifuged at 4000 rpm, at 4 °C for 30 min, and plasma was collected. Plasma ammonia concentrations were measured using a MAK310 Ammonia Assay Kit (Sigma-Aldrich, Saint Louis, MO, USA), according to the manufacturer’s protocol. Plasma aspartate transaminase (AST), alkaline phosphatase (ALP) and bilirubin levels were measured in blood samples collected at the end of the experimental period, using a MAK467 Aspartate Transaminase Assay Kit, a MAK530 Alkaline Phosphatase Assay Kit and a MAK126 Bilirubin Assay Kit (all kits from Sigma-Aldrich), respectively. Plasma haptoglobin was also measured at the end of the experimental period using a Haptoglobin ELISA kit (SEA 817 Ra, MediQip AB, Stockholm, Sweden).

### 4.5. Terminal Procedures

All rats were euthanized using an overdose of sodium pentobarbital (120 mg/kg body weight) upon completion of the experimental period. The rats’ livers were excised and excess connective was removed. The livers were then rinsed in ice-cold saline and blotted dry and weighed. Samples of liver tissue, from the central lobe, were then collected and processed for determination of liver lipid content.

### 4.6. Liver Fat Content Measurement

Lipids were extracted from the liver samples obtained upon euthanasia of the rats, using the Folch method [32,33]. Briefly, liver samples of approximately 150 mg were mechanically homogenized in a chloroform: methanol (2:1) solution for 2 min, using a rotor homogenizer. The samples were then agitated overnight at 25 °C, centrifuged at 3000× *g* for 10 min and the supernatant was collected. Polar lipids were then removed from the samples through the addition of 4 mL of 0.9% NaCl to the supernatant. The mixture was then vortexed for a few seconds and centrifuged at 2500× *g* for 10 min. After discarding the upper phase, the residue remaining in the test tube was rinsed twice with 4 mL of 50% methanol. The bottom (chloroform) phase, containing the fat, was then collected and reduced to dryness under vacuum, using a rotary evaporator and then oven-dried at 45 °C for 2.5 h to remove residual moisture. The fat weight was then determined and the liver fat content was calculated as mg of fat per g of liver tissue.

### 4.7. Statistical Analyses

GraphPad Prism 10.3.1 (San Diego, CA, USA) was used for statistical analyses. Data were tested for normal (Gaussian) distribution using the Shapiro–Wilk normality test. Differences in investigated parameters between dietary groups were assessed using a one-way ANOVA. The effects of hyperammonaemia treatment (Lactulose vs. Rifaximin) and diet (MAD20 vs. MCD), as well as their interaction, was assessed using a two-way ANOVA. The Fisher’s LSD (Least Significant Difference) test was performed as a post hoc test. Differences were considered significant if *p* < 0.05.

## 5. Conclusions and Implications

In conclusion, even though the diets used in the current study caused somewhat similar hyperammonaemia, they seemed to provoke different metabolic consequences. The latter can have an impact on the severity of the resulting hyperammonaemia and thus on the hyperammonaemia-induced encephalopathy. This dietary animal model should be easily reproducible, is cost-effective and does not require any specialist surgical skill/knowledge. The apparent absence of any overt motor and cognitive disturbances (as per our visual observations of the rats), together with specific liver injury, will enable this double animal model to serve as a suitable model to further elucidate the effects of hyperammonaemia of different origins on encephalopathy, as well as for the testing of new, oral therapies for potential HE treatment, in future studies. With regards to the implications from the current study, we would like to suggest that any results obtained from studies involving a single diet-induced HA model may be obscured by the dietary metabolites which may contribute to neurodegeneration. Proposed HA treatments should preferably be studied using a double model to ensure the information obtained concerning the metabolites contributing to the neurodegeneration is reliable.

## Figures and Tables

**Figure 1 ijms-26-01818-f001:**
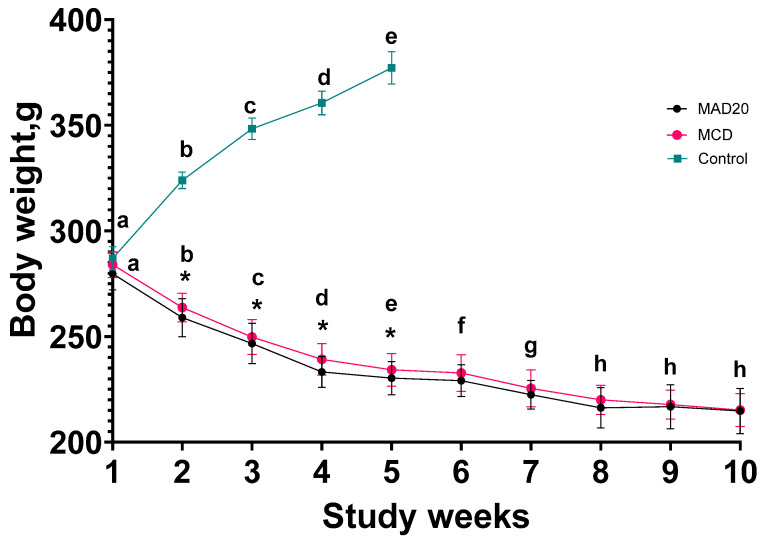
Body mass changes in male, Wistar rats over the 10-week experimental period. Data are presented as Mean ± SD. MCD—rats fed a diet free of methionine and choline, n = 8; MAD20—rats fed a modified MCD with 0.2% choline + 20% ammonium acetate, n = 8; Control—rats fed a regular rat chow diet, n = 4. Different lowercase letters indicate significant differences between body weight measurements at different time points within a single diet group. Asterisks indicate significant differences between diet groups at a given time point. *p* < 0.05 was considered significant.

**Figure 2 ijms-26-01818-f002:**
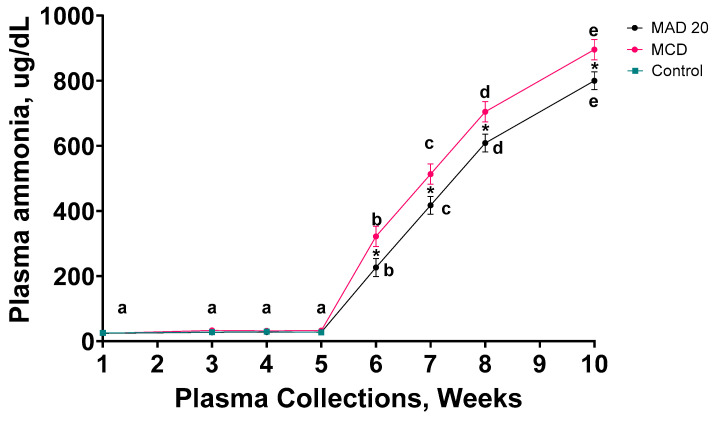
Changes in plasma ammonia levels of male, Wistar rats over the 10-week experimental period. Data are presented as Mean ± SD. MCD—rats fed a diet free of methionine and choline, n = 8; MAD20—rats fed a modified MCD with 0.2% choline + 20% ammonium acetate, n = 8; Control—rats fed a regular rat chow diet, n = 4. Different lowercase letters indicate significant differences when *p* < 0.05 within particular diets, while * indicates significant differences between ammonia levels of different dietary groups at a particular time point.

**Figure 3 ijms-26-01818-f003:**
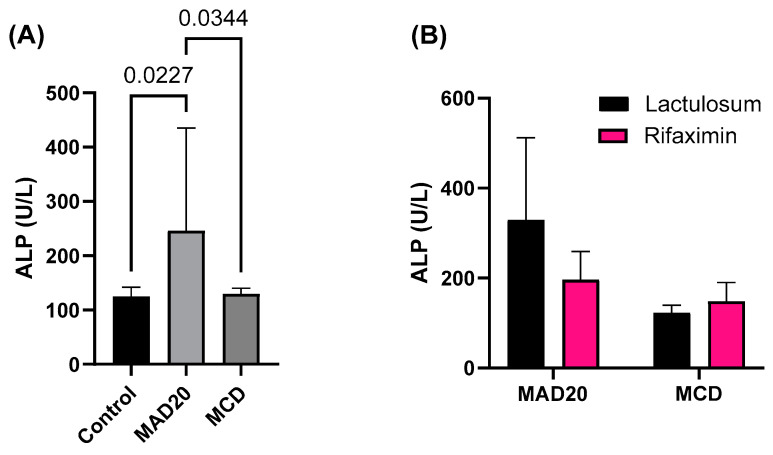
Plasma alkaline phosphatase (ALP) levels of male, Wistar rats at the end of the 10-week experimental period. (**A**) Plasma ALP levels of rats in the different experimental/dietary groups; (**B**) Plasma ALP levels of rats in the MCD and MAD20 groups after receiving either Lactulosum or Rifaximin treatment. Data are presented as Mean ± SD. MCD—rats fed a diet free of methionine and choline, n = 8; MAD20—rats fed a modified MCD with 0.2% choline + 20% ammonium acetate, n = 8; Control—rats fed a regular rat chow diet, n = 4.

**Figure 4 ijms-26-01818-f004:**
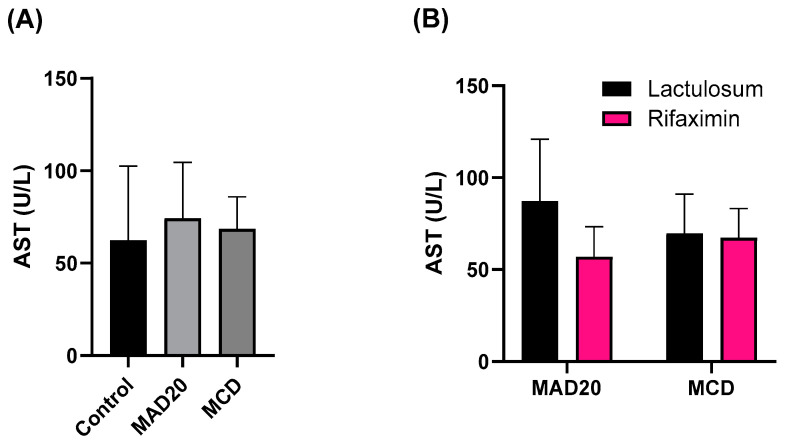
Plasma aspartate aminotransferase (AST) levels of male, Wistar rats at the end of the 10-week experimental period. (**A**) Plasma AST levels of rats in the different experimental/dietary groups; (**B**) Plasma AST levels of rats in the MCD and MAD20 groups after receiving either Lactulosum or Rifaximin treatment. Data are presented as Mean ± SD. MCD—rats fed a diet free of methionine and choline, n = 8; MAD20—rats fed a modified MCD with 0.2% choline + 20% ammonium acetate, n = 8; Control—rats fed a regular rat chow diet, n = 4.

**Figure 5 ijms-26-01818-f005:**
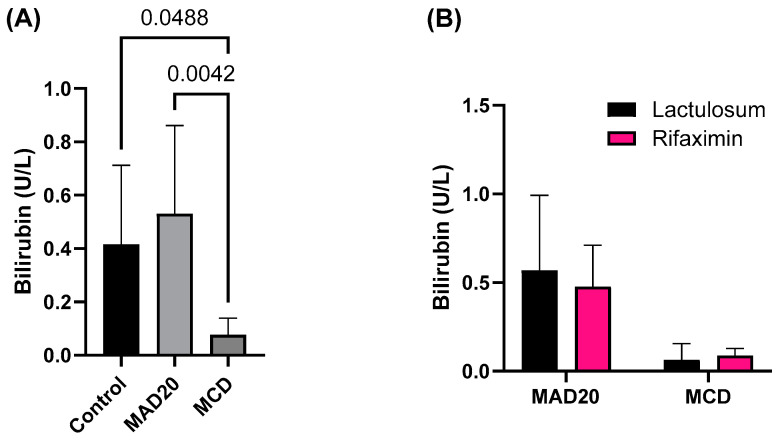
Plasma bilirubin levels of male, Wistar rats at the end of the 10-week experimental period. (**A**) Plasma bilirubin levels of rats in the different experimental/dietary groups; (**B**) Plasma bilirubin levels of rats in the MCD and MAD20 groups after receiving either Lactulosum or Rifaximin treatment. Data are presented as Mean ± SD. MCD—rats fed a diet free of methionine and choline, n = 8; MAD20—rats fed a modified MCD with 0.2% choline + 20% ammonium acetate, n = 8; Control—rats fed a regular rat chow diet, n = 4.

**Figure 6 ijms-26-01818-f006:**
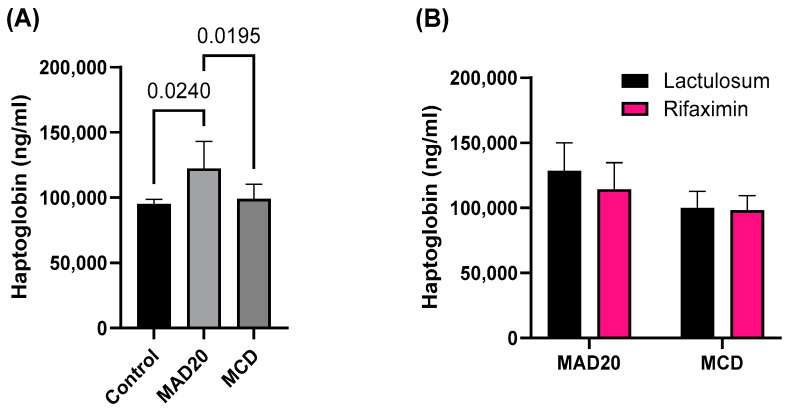
Plasma haptoglobin levels of male Wistar rats at the end of the 10-week experimental period. (**A**) Plasma haptoglobin levels of rats in the different experimental/dietary groups; (**B**) Plasma haptoglobin levels of rats in the MCD and MAD20 groups after receiving either Lactulosum or Rifaximin treatment. Data are presented as Mean ± SD. MCD—rats fed a diet free of methionine and choline, n = 8; MAD20—rats fed a modified MCD with 0.2% choline + 20% ammonium acetate, n = 8; Control—rats fed a regular rat chow diet, n = 4.

**Figure 7 ijms-26-01818-f007:**
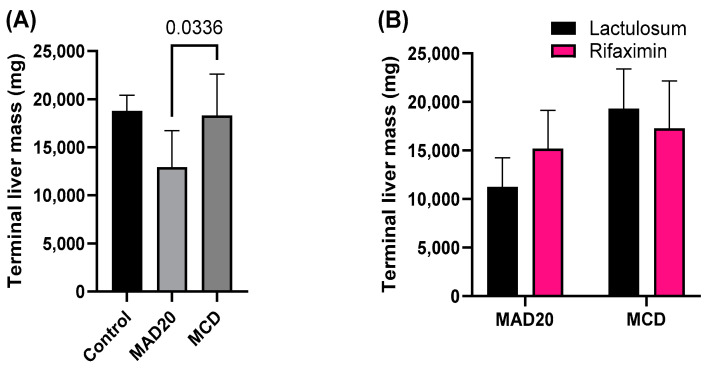
Terminal liver mass of male, Wistar rats at the end of the 10-week experimental period. (**A**) Terminal liver mass of rats in the different experimental/dietary groups; (**B**) Terminal liver mass of rats in the MCD and MAD20 groups after receiving either Lactulosum or Rifaximin treatment. Data are presented as Mean ± SD. MCD—rats fed a diet free of methionine and choline, n = 8; MAD20—rats fed a modified MCD with 0.2% choline + 20% ammonium acetate, n = 8; Control—rats fed a regular rat chow diet, n = 4.

**Figure 8 ijms-26-01818-f008:**
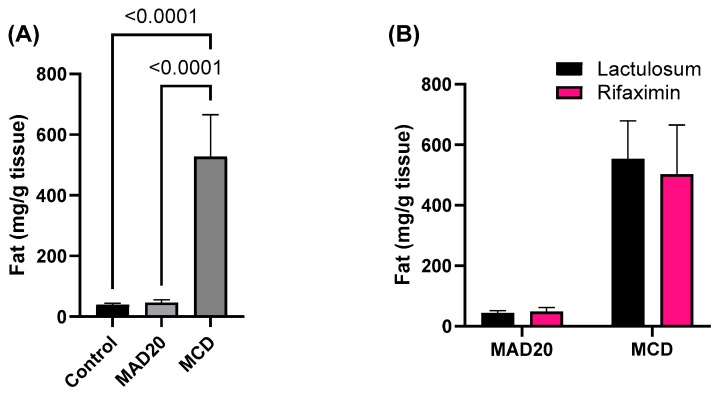
Fat content of the livers of male, Wistar rats at the end of the 10-week experimental period. (**A**) Fat mass of the livers of rats in the different experimental/dietary groups; (**B**) Fat mass of the livers of rats in the MCD and MAD20 groups after receiving either Lactulosum or Rifaximin treatment. Data are presented as Mean ± SD. MCD—rats fed a diet free of methionine and choline, n = 8; MAD20—rats fed a modified MCD with 0.2% choline + 20% ammonium acetate, n = 8; Control—rats fed a regular rat chow diet, n = 4.

**Figure 9 ijms-26-01818-f009:**
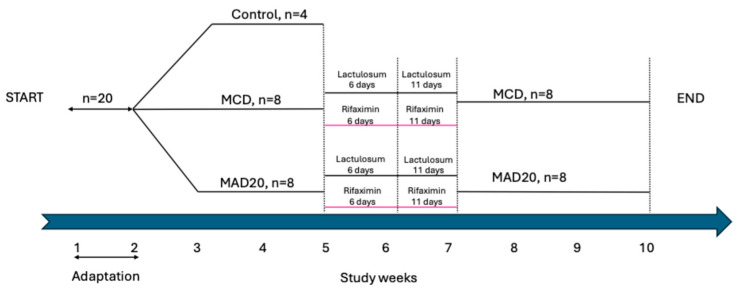
Experimental design (MCD = diet without choline and methionine; MAD20 = diet without methionine, supplemented with 20% ammonia acetate).

**Table 1 ijms-26-01818-t001:** Blood ammonia levels for individual rats during the experimental period.

Diet	Animal Number	Plasma Ammonia Concentration (µg/mL) [at Each Blood Draw]							
		Week 1	2	3	4	5	6	7	8
**MAD20**	7	25.43	28.57	34.96	32.22	188.00	379.25	570.50	761.75
	9	22.83	25.04	24.52	31.96	200.75	392.00	583.25	774.50
	10	21.13	29.35	25.43	26.35	213.50	404.75	596.00	787.25
	12	27.91	30.26	28.30	30.18	239.00	430.25	621.50	812.75
	**Mean**	**24.33**	**28.31**	**28.30**	**30.18**	**210.31**	**401.56**	**592.81**	**784.06**
	**SD**	**2.97**	**2.28**	**4.72**	**2.71**	**21.77**	**21.77**	**21.77**	**21.77**
	11	28.70	22.17	29.61	23.48	226.25	417.50	608.75	800.00
	13	20.22	26.74	26.74	24.00	251.75	443.00	634.25	825.50
	14	21.91	29.74	27.65	29.22	264.50	455.75	647.00	838.25
	22	21.71	24.72	25.58	25.59	201.76	371.76	541.76	711.76
	**Mean**	**23.14**	**25.84**	**27.40**	**25.57**	**236.06**	**422.00**	**607.94**	**793.88**
	**SD**	**3.79**	**3.20**	**1.70**	**2.59**	**27.86**	**37.08**	**46.90**	**57.01**
	**Mean**	**23.73**	**27.07**	**27.85**	**27.87**	**223.19**	**411.78**	**600.38**	**788.97**
	**SD**	**3.21**	**2.89**	**3.32**	**3.48**	**26.93**	**30.20**	**34.80**	**40.29**
**MCD**	15	30.00	36.39	30.26	28.57	277.25	468.50	659.75	851.00
	16	21.65	42.91	34.96	33.78	290.00	481.25	672.50	863.75
	17	23.22	28.57	28.17	30.78	302.75	494.00	685.25	876.50
	18	28.96	28.83	32.87	34.30	315.50	506.75	698.00	889.25
	**Mean**	**25.96**	**34.18**	**31.57**	**31.86**	**296.38**	**487.63**	**678.88**	**870.13**
	**SD**	**4.14**	**6.86**	**2.97**	**2.69**	**16.46**	**16.46**	**16.46**	**16.46**
	19	20.09	22.43	43.57	49.30	328.25	519.50	710.75	902.00
	20	24.91	33.65	26.61	28.04	341.00	532.25	723.50	914.75
	6	16.30	38.87	31.43	28.17	353.75	545.00	736.25	927.50
	21	26.22	34.04	26.74	29.61	366.50	557.75	749.00	940.25
	**Mean**	**21.88**	**32.25**	**32.09**	**33.78**	**347.38**	**538.63**	**729.88**	**921.13**
	**SD**	**4.56**	**6.96**	**7.98**	**10.37**	**16.46**	**16.46**	**16.46**	**16.46**
	**Mean**	**23.92**	**33.21**	**31.83**	**32.82**	**321.88**	**513.13**	**704.38**	**895.63**
	**SD**	**4.58**	**6.48**	**5.58**	**7.09**	**31.23**	**31.23**	**31.23**	**31.23**
**Control**	1	26.61	27.39	23.35	24.52	-	-	-	-
	2	22.70	27.26	38.09	30.18	-	-	-	-
	3	26.09	26.61	30.00	27.78	-	-	-	-
	4	27.00	31.43	25.30	29.61	-	-	-	-
	**Mean**	**25.60**	**28.17**	**29.18**	**28.02**	**-**	**-**	**-**	**-**
	**SD**	**1.97**	**2.20**	**6.56**	**2.55**	**-**	**-**	**-**	**-**

Values with grey background were obtained during Lactulose treatment, while values with pink background were obtained during Rifaximin treatment.

**Table 2 ijms-26-01818-t002:** Ingredients and proximate composition of the experimental diets.

Ingredient	MCD	MAD + 20% Ammonia	Control (Standard Diet)
**Proximate contents (%)**			
Crude protein	15.0 ^(1)^	15.0 ^(1)^	17.0
Crude fat	10.0	10.0	10.0
Crude fibre	3.0	3.0	5.0
Crude ash	3.3	3.3	3.3
Ammonium acetate		20	
Starch	19.2	14.4	20
Sugar	45.9	30.8	45
Lysine	1.40	1.40	
Methionine			
Cystine	0.35	0.35	
Threonine	0.80	0.80	
Tryptophan	0.18	0.18	
**Choline mg/kg**		860	
**Energy ^(2)^ MJ/kg**	17.7	14.4	17.6

^(1)^ Protein calculated from the N content of the amino acids (N × 6.25). The nitrogen of ammonia acetate would theoretically increase the protein content to 37.6% (N × 6.25). ^(2)^ Energy content of ammonia acetate not considered.

## Data Availability

The raw data supporting the conclusions of this article will be made available by the authors on request.

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
