# Peer review of "Rat Model of Endogenous and Exogenous Hyperammonaemia Induced by Different Diets"

_ijms, 2025, doi:10.3390/ijms26051818_

Round 1

Reviewer 1 Report

Comments and Suggestions for Authors

The article titled “Rat model of endogenous and exogenous hyperammonaemia induced by different diets”. It is recommended to reject the manuscript, or submit it again after adding the molecular mechanism.This research is too superficial and not specific or in-depth enough.

The specific content is as follows:

1. What is the specific molecular mechanism by which MCD and MAD cause very high plasma ammonia levels?The article does not provide a specific molecular mechanism that may lead to increased ammonia levels, and the molecular mechanism is not solid.

2. What is the main molecular mechanism in HA that leads to decreased bilirubin levels and increased liver fat? There is no clear explanation for the effect of increased ammonia levels on bilirubin and liver fat.

3. Why didn't the dietitians who studied MAD + 20% ammonium acetate do the MCD group? This is inconsistent with previous studies, and the specific molecular mechanism of reducing the levels of two proteins (find it in the article and add it in) and liver quality is not clearly explained.

4. The n number of all the figures in the article is not included. Please make sure to add the n number of all statistical graphs when publishing the article later.

5. The molecular mechanism is missing.

6. Conclusions: The authors should add a conclusion section to summarize this item.

7. Abbreviations: Authors do not need to list abbreviations separately; the full name (abbreviation) should be used when it first appears in the main text of the article.

8. References: Authors should revise the reference format according to the journal and remove references that are too old. The author should add more references within the past five years.

Reviewer 2 Report

Comments and Suggestions for Authors

I enjoyed reading this manuscript. Althouugh this paper is well written. There are major areas where the authors need to improve.  I am including suggestions for improvement

1. results: the control group figures should be incoporated into the figure and not shown seperately. This ensures proper comparison between controls and experimental models

2. Blood draw periods or intervals should be clearly shown/indicated in table 1. Avoid interpreting results beyond what is shown or beyond the statistics provided in the table ore figures of results

3. Discussion of results should be mainly based on significant results

Round 2

Reviewer 1 Report

Comments and Suggestions for Authors

Since the authors have responded positively to the reviewers' comments and made improvements in the revised manuscript, the article can be considered for acceptance. However, it is recommended that the authors further add a brief discussion of the molecular mechanism in the final version, even if it is just to propose the direction of future research, to enhance the completeness and scientificity of the article. In addition, the authors are requested to elaborate on the limitations of the study in more detail in the Discussion section.
